# Ursodeoxycholic acid as a novel disease-modifying treatment for Parkinson's disease: protocol for a two-centre, randomised, double-blind, placebo-controlled trial, The 'UP' study

Thomas Payne [1,2] Matilde Sassani,[1] Ellen Buckley [2,3] Sarah Moll,[2] Adriana Anton,[2,4] Matthew Appleby,[5] Seema Maru,[5] Rosie Taylor,[6] Alisdair McNeill,[1] N Hoggard,[4] Claudia Mazza,[3] Iain D Wilkinson,[4] Thomas Jenkins,[1] Thomas Foltynie,[5] O Bandmann [1,2]

For numbered affiliations see end of article.

**Correspondence to**
Dr O Bandmann;
o.bandmann@sheffield.ac.uk

## ABSTRACT

**Introduction** There are no disease-modifying treatments for Parkinson's disease (PD). We undertook the first drug screen in PD patient tissue and idntified ursodeoxycholic acid (UDCA) as a promising mitochondrial rescue agent. The aims of this trial are to determine safety and tolerability of UDCA in PD at 30 mg/kg, confirm the target engagement of UDCA, apply a novel motion sensor-based approach to quantify disease progression objectively, and estimate the mean effect size and its variance on the change in motor severity.

**Methods and analysis** This is a phase II, two-centre, double-blind, randomised, placebo-controlled trial of UDCA at a dose of 30 mg/kg in 30 participants with early PD. Treatment duration is 48 weeks, followed by an 8-week washout phase. Randomisation is 2:1, drug to placebo. Assessments are performed at baseline, week 12, 24, 36, 48 and 56. The primary outcome is safety and tolerability. Secondary outcomes will compare the change between baseline and week 48 using the following three approaches: the Movement Disorders Society Unified Parkinson's Disease Rating Scale Part 3 in the practically defined 'OFF' medication state; confirmation of target engagement, applying [31]Phosphorus MR Spectroscopy to assess the levels of ATP and relevant metabolites in the brain; and objective quantification of motor impairment, using a validated, motion sensor-based approach. The primary outcome will be reported using descriptive statistics and comparisons between treatment groups. For each secondary outcome, the change from baseline will be summarised within treatment groups using summary statistics and appropriate statistical tests assessing for significant differences. All outcomes will use an intention-to-treat analysis population.

**Ethics and dissemination** This trial has been approved by the East of England – Cambridgeshire and Hertfordshire Research Ethics committee. Results will be disseminated in peer-reviewed journals, presentations at scientific meetings and to patients in a lay-summary format.

### Strengths and limitations of this study

► This is the first double-blind, randomised, placebo-controlled trial of ursodeoxycholic acid (UDCA) in Parkinson's disease (PD).
► This study uses novel secondary outcomes not previously used in a clinical trial studying PD; namely [31]Phosphorus MR spectroscopy ([31]P-MRS) of disease-specific regions and detailed, complementary home and clinic-based motor activity and gait analysis.
► [31]P-MRS will allow the assessment of mitochondrial dysfunction directly in the midbrain, including the substantia nigra, the most severely affected brain area in PD.
► A limitation of the study is the considerable number of capsules patients will have to take; patients will on average be taking an additional nine extra capsules of medication each day through the trial, significantly increasing their 'pill burden'.
► A further limitation is the small sample size of n=30 with 20 patients on UDCA and 10 patients on placebo, it will not be possible to draw firm conclusions about the neuroprotective effect of UDCA in PD but will allow for appropriate power and sample size calculations for future studies.

**Trial registration number** NCT03840005.

## INTRODUCTION

Parkinson's disease (PD) is a progressive neurodegenerative disorder comprising gait impairment, bradykinesia, rigidity and tremor.[1] It is the second most common neurodegenerative disorder, predicted to double in global prevalence between 2005 and 2030.[2] Developing disease-modifying therapies is a crucial step in reducing the

associated morbidity of PD and to delay the development of late-stage complications such as dementia, postural instability and psychosis.

Mitochondrial dysfunction is a key pathogenic mechanism in both sporadic and familial PD and therefore a promising target for disease-modifying therapy.[3] Our group undertook the first drug screen in genetically stratified PD patient tissue.[4 5] This approach identified ursodeoxycholic acid (UDCA) as a particularly promising mitochondrial rescue compound.[5] Other groups demonstrated independently the neuroprotective effect of UDCA and its taurine conjugate TUDCA in the 1-methyl-4-phenyl-1,2,3,6-tetrahydropyridine (MPTP) mouse model and the rotenone rat model of PD.[6 7]

The mode of action of UDCA remains to be fully elucidated. Current literature would suggest that it appears to be Akt mediated. Both ursocholanic acid and TUDCA have been demonstrated to induce Akt phosphorylation.[4 7] Akt activation requires phosphorylation at two sites and promotes cell survival through several mechanisms, failure of activation is a common finding underlying neurodegeneration.[4] Reduced Akt signalling has been found in in-vitro models of PD and in sporadic PD brains postmortem in the substantia nigra.[8 9]

UDCA has been in clinical use for decades primarily for primary biliary cholangitis (previously primary biliary cirrhosis) with excellent safety and tolerability at the standard dose of 15 mg/kg.[10] UDCA has also been well tolerated at a higher dose of 30 mg/kg over 2 years in patients with primary sclerosing cholangitis.[11] UDCA is a naturally occurring bile acid but normally only forms 1%–3% of total endogenous human bile acids. However, in patients on standard therapeutic doses of UDCA (13–15 mg/kg/day), UDCA may form up to 40% of total bile acids. Intestinal absorption after an oral dose is high with a first-pass clearance of about 50%–60%. Plasma levels reach maximum concentrations after 60 min after ingestion with another peak at 3 hours.[12]

A pharmacokinetic study of UDCA in Motor Neurone Disease (MND) demonstrated a significant correlation between serum concentration at 1-hour postdose and CSF concentration 2 hours postdose, with most of the variability in CSF concentrations (78%) explained by variability in serum concentrations. Mean CSF concentration postdose at 15 mg/kg was 86.69 nmol/L, at 30 mg/kg was 114.22 nmol/L and 50 mg/kg was 191.11 nmol/L.[13]

The main objectives of this trial (The UP Study) are to demonstrate the safety and tolerability of UDCA in PD at a dose of 30 mg/kg and to explore the effects of UDCA on novel outcome measures such as [31]Phosphorus MR spectroscopy ([31]P-MRS) and the objective quantification of motor impairment, using a sensor-based approach. Additionally, we hope to collect an estimate of the effect size and variance of UDCA on the change in motor severity of PD over 1 year compared with placebo using long-established clinical assessment tools.

## METHODS AND ANALYSIS
### Design
This is a phase II, two-centre, double-blind, randomised, placebo-controlled trial of 30 mg/kg of UDCA in early PD. Treatment duration with drug or placebo is 48 weeks in total, followed by an 8-week washout phase. Thirty participants will be included. Randomisation is 2:1 in favour of drug to placebo. The choice of 30 mg/kg day has been informed by previous pharmacokinetic studies in MND, this dose allows effective penetrance of the central nervous system but also balances the exposure to a potentially higher risk of side effects with increasing doses and possible issues with compliance due to the then very large number of additional tablets the patients would need to take.[13]

### Participants
Patients with early PD, as defined by a clinical diagnosis made by a movement disorders specialist according to the Queen Square Brain Bank Criteria within 3 years prior to recruitment and who demonstrate a clear subjective response to dopaminergic medication, confirmed by the treating physician, will be recruited from two sites; Sheffield Teaching Hospitals National Health Service (NHS) Trust (STH) and University College London Hospitals NHS Foundation Trust (UCLH). Key inclusion and exclusion criteria can be found in box 1.[14]

Participants are typically recruited through specialist movement disorders clinics at both trial sites. The trial has also been advertised online by the Parkinson's UK website, the Cure Parkinson's Trust, the Sheffield National Institute for Health-Related Research (NIHR)-Biomedical Research Centre website and the NIHR Clinical Research Network websites. Trial advertisements direct participants to contact the STH study team to be provided with a patient information sheet and a reply slip to confirm ongoing interest and to organise a pre-screening telephone call to confirm eligibility and suitability for the study.

Study visits either take place at the clinical research facility (CRF) of the Royal Hallamshire Hospital, Sheffield, for STH participants or at the Leonard Wolfson Experimental Neurology Centre, Queen Square, London for UCLH participants.

### Primary outcome
The primary outcome for the UP study is to compare the safety and tolerability of UDCA at 30 mg/kg in PD compared with placebo as indicated by the following: the number of serious adverse events (SAEs), number of adverse treatment reactions and the number of patients completing the study. The safety and tolerability of UDCA in this study will be compared descriptively with the reported safety and tolerability of exenatide in the

## Box 1  Key inclusion and exclusion criteria for the UP study

### Key inclusion criteria

► Diagnosis of Parkinson's disease (PD) ≤3 years ago based on Queen Square Brain Bank criteria.[14]
► Subjective improvement of motor impairment on dopaminergic medication with confirmation by a movement disorders expert.
► Hoehn and Yahr stage ≤2.5 in the practically defined 'ON' medication state.
► Age 18–75 years of any gender.
► Able to comply with study protocol and willing to attend necessary study visits.
► Ability to communicate in English.
► Ability to take study drug.

### Key exclusion criteria

► Diagnosis or suspicion of other cause of parkinsonism.
► Known abnormality on CT or MRI brain imaging considered likely to compromise compliance with $^{31}$Phosphorus MR Spectroscopy acquisition.
► Known claustrophobia or other reasons why patient could not tolerate or be suitable for MRI.
► Current or previous exposure to ursodeoxycholic acid.
► Current or previous diagnosis of liver disease (including biliary obstruction), in particular primary biliary cirrhosis judged to be significant.
► Prior intracerebral surgical intervention for PD (including deep-brain stimulation).
► Already actively participating in a trial of a device, drug or surgical treatment for PD.
► Participants who lack the capacity to give informed consent.
► History of alcoholism.
► Women of childbearing potential or pregnancy.
► Concurrent severe depression defined by a score >16 on the Montgomery-Asberg Depression Rating Scale.
► Concurrent dementia defined by a score lower than 25 on the Montreal Cognitive Assessment.
► Any medical or psychiatric condition which in the investigator's opinion compromises the potential participant's ability to participate.
► Serum transaminases more than two times upper limit of normal.
► Patients on ciclosporin, nitrendipine or dapsone.
► Participants with previous or current diagnosis of inflammatory bowel disease.

exenatide-PD trial, which followed a broadly similar trial design.[15]

### Secondary outcomes

The effect of UDCA versus placebo will be assessed as a change from baseline to week 48 for the following secondary outcomes:

1. Clinical assessment using the Movement Disorders Society Unified Parkinson's Disease Rating Scale (MDS-UPDRS) Part 3 motor examination in the practically-defined 'OFF' medication state.
2. In vivo measures of high and low energy metabolite levels (including adenosine triphosphate (ATP), phosphocreatine (PCr) and inorganic phosphate (Pi)) derived from multi-voxel brain $^{31}$P-MRS at baseline and week 48.

3. Sensor-based, objective quantification of motor impairment using data collected with wearable sensors both in supervised (OptoGait and Opals systems, Sheffield patients only, Dynaport Movemonitor+, all patients) as well as in unsupervised real-life conditions (Dynaport Movemonitor+, all patients).

### Screening visit

Participants likely to be eligible will be invited for a screening visit where all inclusion and exclusion criteria will be reviewed. Participants will be offered the opportunity to discuss the trial and have all questions answered after which they will be asked to provide written informed consent before proceeding to further assessment. Participants will have a full demographic, medical and concomitant medication history taken and reviewed. A physical examination to confirm the diagnosis of PD and exclude PD 'mimic' conditions will be performed. A Montreal Cognitive Assessment (MoCA) and Montgomery-Asberg Depression Rating Scale (MADRS) will be performed to exclude concurrent dementia or severe active depression.[16 17] Safety bloods (full blood count, urea and electrolytes, liver function tests, blood glucose, Haemoglobin A1C (HbA1C), lipid profile) and an ECG will be performed at the screening visit. If the participant remains eligible, they will be provided with an activity monitor (McRoberts, Dynaport MoveMonitor+) to wear for 1 week prior to the baseline visit as described later. For those undergoing $^{31}$P-MRS, this will be arranged within 1 week before or on the day of the baseline visit, as described later. The baseline visit will be completed within 8 weeks of screening.

### Baseline visit, randomisation and blinding

Randomisation to either active compound or placebo will be administered using a centralised, web-based system hosted by epiGenesys (a wholly owned subsidiary of the University of Sheffield) on behalf of the University of Sheffield Clinical Trials Research Unit (CTRU).

MDS-UPDRS part 3 Motor Examination is performed in the 'OFF' state.[18] The practically defined 'OFF' state in this study requires participants to not have taken medication for 8 hours in the case of any drug containing Levodopa, or at least 36 hours in the case of longer-acting agents such as dopamine agonists or enzyme inhibitors.

The supervised gait analysis is performed using a combination of an instrumented photoelectric walkway system (Microgate, OptoGait) and inertial sensors (APDM, Opal) system as described below.

Participants will then be invited to take their usual dopaminergic medication and after a minimum of 60 min undergo the following procedures to reassess them in the practically defined 'ON' state: MDS-UPDRS Parts 1–4 I in the 'ON' state, Non-Motor Symptom Questionnaire (NMS-QUEST) and The 39-Item Parkinson's Disease Questionnaire (PDQ-39).[18–20]

## Intervention

All study medication is provided as a white powder in a hard clear gelatine capsule. Placebo and study drug are completely matched with no identifiable differences in taste, appearance or smell. All packaging and labelling is identical. Each capsule of the active drug contains 250 mg of UDCA.

Treatment with UDCA is started at a dose of 250 mg (one capsule) per day with an increase by 250 mg every 3 days until the target dose is reached, which is divided into three doses.[21] Most patients are expected to reach their target dose within 3–4 weeks and be on 9–10 capsules per day.

All participants, trial management and medical staff will be blinded to treatment. Participants undergo clinical assessments by the same blinded assessor at each site who is not involved with safety, AE monitoring or dose titration to avoid any assessment bias or accidental unblinding.

## Assessment procedures

Following randomisation and baseline visit, a total of five further visits will be completed at week 12, 24, 36, 48 and 56. At week 48, treatment is completed and all medication returned. A final visit at week 56 for final safety monitoring and outcome measurement completes the study. Week 12 and 36 visits are purely for safety monitoring and medication supply.

The MDS-UPDRS part 3 is completed in the practically defined 'ON' state at week 24 and in the 'OFF' state at week 48 and 56. The complete MDS-UPDRS (Parts 1–4) is completed in the 'ON' state at week 48 and 56.

The $^{31}$P-MRS is repeated in the 7 days prior to week 48 for UCLH participants and on the day of the week 48 visit for STH participants. The week-long unsupervised at-home physical activity monitoring (PAM) is repeated in the 7 days prior to week 48.

The MoCA, NMS-QUEST, PDQ-39 and MADRS are repeated at week 48 and 56.

At each visit, safety bloods (full blood count, urea and electrolytes, liver function tests, blood glucose, HbA1C, lipid profile) will be obtained. In addition, at each visit a 20 mL serum sample is taken for long-term storage and future research. At the baseline visit, blood is taken for genetic analysis, this will be performed using the Neuro-Chip Assay that assesses for approximately 180 000 genetic variants associated with neurological diseases.[22]

A full schedule of activities can be seen in table 1.

## Exploratory outcomes

The exploratory outcomes will consist of the change between week 48 and 56 in the following: MDS-UPDRS part 3 'OFF' scores, complete MDS-UPDRS (parts 1–4) 'ON' scores, total Levodopa equivalent daily dose, MoCA, MADRS, NMS-QUEST and PDQ-39.

The repeat assessments at week 56 (8 weeks after cessation of the study medication) will help to determine whether there is a sustained effect of UDCA on both motor and non-motor aspects of PD which would be in keeping with the assumption of a neuroprotective effect. Conversely, a rapid deterioration of these clinical parameters after cessation of the study drug would suggest a symptomatic effect of UDCA.

As an additional variable to be used in exploratory analysis a validated prognostic model calculating the risk of progression to an unfavourable outcome (either postural instability or dementia at 5 years) will be applied to each participant.[23] We hope that this variable will account for some of the inherent heterogeneity among participants for their speed of clinical progression.

## Sample size

The primary outcome of interest for this study is the safety and tolerability of UDCA which will be assessed by comparing the rate of SAEs in the UDCA and placebo groups, alongside review of adverse treatment reactions and study completion. As the study is a pilot, it is not powered to compare the SAE rate between the groups statistically, but any SAEs in either group will be presented descriptively, the placebo group providing a baseline against which to view any SAEs in the UDCA group. Should this study result in no SAEs then it would be of interest to determine how likely it is that a larger study would find an intolerable rate of SAEs. For this purpose, we will consider the rate of SAEs reported in the exenatide PD trial to be tolerable and acceptable (ie, 20%).[15] In this study, should no SAEs be found in the group receiving UDCA (n=20) then the likelihood that the true SAE rate is less than 20% is 0.990778.

The sample size has not been prospectively adjusted to account for any lost to follow-up. Instead, as the trial is of a relatively short duration we have instead allowed for any participants withdrawing from the study or lost to follow-up before the completion of 12 weeks of treatment to be replaced with a new participant.

The study has not been powered formally for the secondary or exploratory outcome measures, therefore, interpretation will concentrate on observed trends and confidence intervals for estimated differences. The data collected for the secondary and exploratory outcomes will allow the estimation of the effect size and variance in each outcome to facilitate formal power calculations for future phase III studies. Of note, there is currently no longitudinal clinical trial data using either $^{31}$P-MRS or our sensor-based approached quantification of motor impairment. The collection of such data is critical to allow high-quality future trial design using these novel outcome measures.

## Patient and public involvement

Patient representatives have been involved in the design of the study protocol and have contributed to the generation of participant facing study documentation. Recruitment to the study will be aided by both local PD groups and publicised by The Cure Parkinson's Trust, Parkinson's

**Table 1** Schedule of activities for the UP study

| | Procedure | Screening | Baseline | Week 12 | Week 24 | Week 36 | Week 48 | Week 56 |
|---|---|---|---|---|---|---|---|---|
| Medical history | Consent | X | | | | | | |
| | Review inclusion/exclusion criteria | X | X | | | | | |
| | Demographics | X | | | | | | |
| | Medical history and physical examination | X | | | | | | |
| | Height and weight | X | | | | | X | |
| | Genetics Sample | | X | | | | | |
| Medication | Randomisation | | X | | | | | |
| | Medication supply | | X | X | X | X | | |
| | Concomitant medication review | X | X | X | X | X | X | X |
| | Compliance review | | | X | X | X | X | |
| Clinical assessment/ outcome measures | MDS-UPDRS Part 3 'OFF' | | X | | | | X | X |
| | MDS-UPDRS Part 3 'ON' | | | | X | | | |
| | MDS-UPDRS Parts 1–4 'ON' | | X | | | | X | X |
| | MoCA, MADRS | X | | | | | X | X |
| | PDQ-39 | | X | | | | X | X |
| | NMS-QUEST | | X | | | | X | |
| Sensor-based analysis | Dynaport Move Monitor +7 days recording | X | | | | | X (7 days prior) | |
| | OptoGait/Opals gait assessment 'OFF' | | X | | | | X | |
| MRI | $^{31}$P-MRS | | X | | | | X | |
| Safety monitoring | Safety bloods | X | X | X | X | X | X | X |
| | ECG | X | | | X | | | |
| | AE review | | X | X | X | X | X | X |

AE, adverse event; MADRS, Montgomery-Asberg Depression Rating Scale; MDS-UPDRS, Movement Disorders Society Unified Parkinson's Disease Rating Scale; MoCA, Montreal Cognitive Assessment; NMS-QUEST, Non-Motor Symptom Questionnaire; PDQ-39, 39-Item Parkinson's Disease Questionnaire; $^{31}$P-MRS, $^{31}$Phosphorus MR spectroscopy.

UK and Michael J Fox Foundation. Results will be disseminated to all participants on completion of the trial.

## OUTCOME MEASURES
### Safety monitoring
At each visit, participants are asked to report any AEs that have occurred since the previous visit. AEs may also be detected by the study team reviewing the patient or through notification by the participant's primary care physician. All AEs are assessed by a study doctor for their severity, likely relationship to study drug and required action by a study doctor not involved in the blinded assessment of the patient. All SAEs will be recorded and reported to the sponsor regardless of relation to trial treatment within 24 hours. Any suspected unexpected serious adverse reactions will be reported to the sponsor immediately to allow facilitation of unblinding as necessary. All AEs reported will be reviewed by the trial management group, trial steering group and monitored by an independent data monitoring committee.

Unblinding requests from other clinicians responsible for a patient's care will be handled by the principal investigator (PI) at each site. The PI at each site may also choose to unblind in response to reported AEs as they are reported.

In the event that side effects such as diarrhoea do not resolve and become persistent or intolerable then the patient can have their dose adjusted to their last tolerated dose for the remainder of the study.

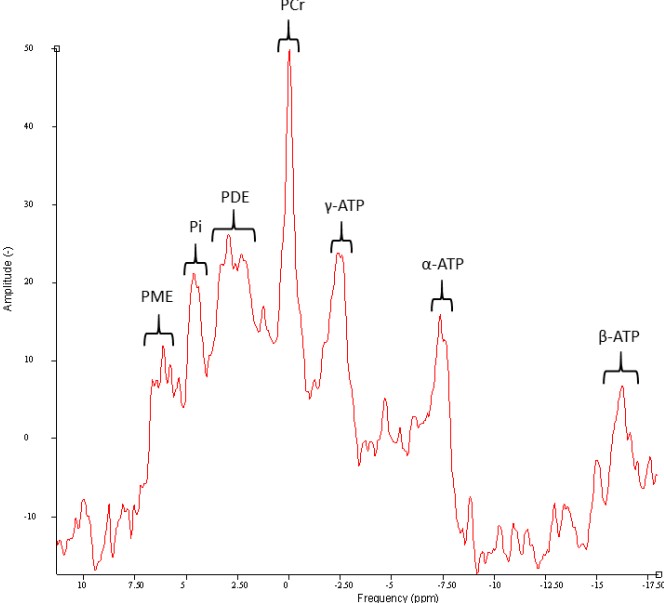

**Figure 1** Representative 31P-MRS spectra obtained from the midbrain of a healthy volunteer following appropriate phasing and 10 Hz Lorentzian apodisation with no further editing. From left to right, phosphomonoesters (PME), inorganic phosphate (PI), phosphodiesters (PDE), phosphocreatine (PCr) and the three spectral resonances of adenosine triphosphate (γ-,α-,β-ATP). 31P-MRS, 31phosphorus MR spectroscopy.

All participants will be asked to return unused medication, this medication will be counted and recorded to assess compliance.

### Motor measures

The MDS-UPDRS is currently the most used and validated clinical tool to quantify the disease state of an individual with PD.[18] The minimal clinically important difference in the MDS-UPDRS part 3 is reported to be an improvement of 3.25 points for detecting minimal, but clinically pertinent, improvement and a deterioration of 4.63 points for observing minimal, but clinically pertinent, worsening.[24] Over a period of 5 years, MDS-UPDRS part 3 scores were observed to increase (deteriorate) by 2.4 points per year.[25] However, despite expected annual deterioration being well characterised, rate of decline may still depend on disease stage and therefore contemporaneous placebo control data remains essential to evaluate potential new therapies.

### Neuropsychological measures

The MoCA is a globally used and validated measure of cognitive impairment and has been used a broad range of neurological diseases and study designs.[16] The MADRS has been validated in PD as a screening tool for major depression.[17 26]

### Non-motor and quality of life measures

NMS-QUEST is a clinical screening tool that covers a wide range of non-motor symptoms.[20] PDQ-39 is a validated and widely used quality of life questionnaire that

covers a range of measures such as emotional well-being, activities of daily living and mobility in the context of PD.[19] The total equivalent levodopa dose is calculated using calculations and equivalencies generated previously in a systematic review and allows quantitative comparisons between patients on different medication regimes.[27]

### 31Phosphorous MR spectroscopy

31P-MRS is experienced by the patient in the same manner as a standard clinical MRI scan. As the metabolites of interest are phosphorus based, it provides the opportunity to investigate key metabolites in bioenergetics such as ATP, PCr and Pi which all have clear spectroscopic resonances (figure 1). It is, therefore, an ideal approach to assess mitochondrial function in vivo. Ratio measures such as Pi/ATP and PCr/ATP have been shown to reflect the status of different aspects of oxidative phosphorylation pathways.[28]

Two-dimensional chemical shift imaging (CSI) with image-selected in vivo spectroscopy will be used for spectral spatial localisation,[29 30] with a dedicated multinuclear MRI system (Ingenia 3.0T, Philips Healthcare, Best, NL) and dual-tuned 1H/31P head coil (Rapid Biomedical, Würzburg, Germany). Standard clinical T1 and T2-weighted imaging will allow the alignment of the two 31P axial CSI sequences as shown in figure 2. The two sequences will be aligned to obtain spectra from both the putamen (voxels for both anterior and posterior putamen bilaterally) and the midbrain (one voxel for each left and right). This is a clear advantage over alternative techniques that typically use surface coils as it allows the localisation of spectra to these specific brain regions typically involved in early PD. Imaging both anatomical regions is of importance as one mechanism of mitochondrial dysfunction in PD may be that of retrograde axonal degeneration, therefore, spectra from the striatum may show clear mitochondrial dysfunction even in early disease independent of findings in the midbrain. Previous cross-sectional work using a similar 31P-MRS protocol has demonstrated reductions in ATP and PCr in PD compared with controls in both the putamen and midbrain.[31] Additionally, a further study demonstrated that Pi/ATP ratios were increased in PD compared with controls.[32]

Details of the acquisition sequences are shown in table 2. Spectra will be processed in the time domain using jMRUI software V.5.2 (http://www.jmrui.eu) and the Advanced Method for Accurate, Robust and Efficient Spectral fitting (AMARES) algorithm is used to determine the relative area under each peak.[33–35] Analysis of the 31P-MRS data will focus on the change between randomisation and week 48 of normalised amplitudes of ATP, PCr and Pi, and ratio values such as PCr/ATP and Pi/ATP that assess bioenergetic dysfunction. All STH patients will undergo 31P-MRS. UCLH patients are also invited to attend the STH site for 31P-MRS.

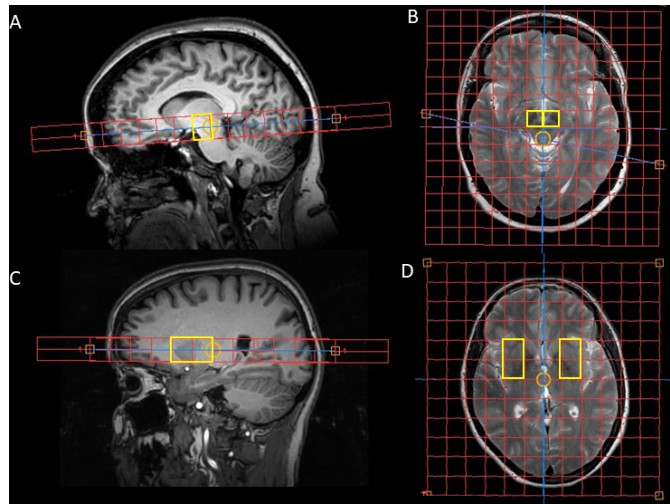

**Figure 2** The substantia nigra slice is placed to cover the midbrain with the highlighted voxels of interest for subsequent analyses highlighted in yellow in the sagittal (A) and axial planes (B). Placement of $^{31}$P-MRS slices. The basal ganglia slice is placed over the putamen aligned in both the coronal (C) axial planes (D), and voxels of interest for subsequent analyses are highlighted in yellow. One voxel covers the anterior putamen and another the posterior putamen.

## Gait analysis and activity monitoring

Physical activity and gait capacity will be assessed at two time points, namely prior to/during the baseline visit and prior to/during the week 48 visit at the end of the treatment period.

Physical activity will be assessed using home-based 'real-life' monitoring for seven consecutive days. A lightweight PAM containing a triaxial accelerometer, gyroscope, digital memory card and a battery (McRoberts, Dynaport Movemonitor+Netherlands) has been selected for continuous monitoring in all participants. Participants will wear the device for seven consecutive days and complete a diary to quantify their physical activity and gait characteristics within their normal weekly routine in a 'real-world' setting.

Gait capacity will be assessed during the study visits (figure 3) using a combination of wearable inertial sensors and an instrumented walkway. In particular, participants

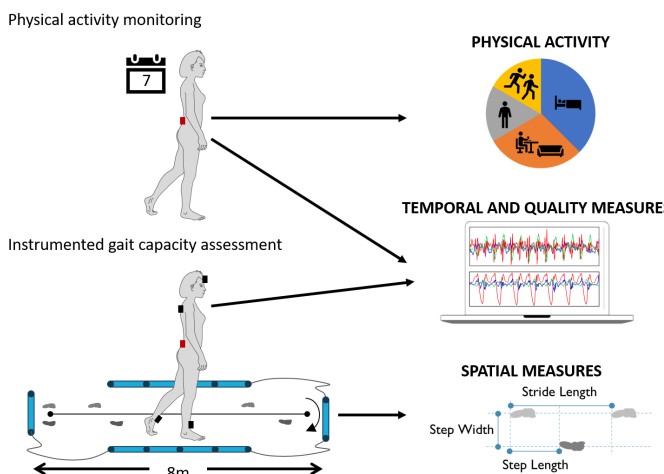

**Figure 3** Protocols deployed at the two sites. all participants undergo 7-day physical activity monitoring in order to estimate physical activity levels and capture temporal and gait quality measures in a real-world setting. In-clinic instrumented gait tasks are also completed at both sites to provide spatiotemporal and gait quality measures of gait capacity. At UCL only red sensor location is implemented.

will complete gait analysis tasks during baseline and week 48 at the respective centre's CRFs (STH and UCLH). Patients will complete three short gait tasks. First, participants will be asked to complete the 3 m timed up and go test walk at self-selected speed. It is an assessment of functional mobility that incorporates transitional actions of standing, turning and sitting.[36 37] Then participants will complete two continuous gait tasks at self-selected preferred, and fast paced walking speeds. Each trial will consist of walking back and forth at least six times along the 8 m walkway with periods of quiet standing recorded at the start and end of each trial. At both sites, participants will wear the Dynaport Movemonitor+during instrumented gait tasks. At the Sheffield site, an instrumented 8 m walkway (OptoGait, Microgate Corporation, Bolzano, Italy) and a set of inertial sensors (Opals, APDM, Portland, Oregon, USA) will also be implemented. The instrumented walkway uses bar-mounted LEDs in a two dimensional configuration. The infrared signals transmitted are broken by the movement of the research

**Table 2** Detailed parameters of the $^{31}$P protocol for acquisition

| Sequence description | Localisation | Decoupling, NOE | TR (ms) | TE (ms) | NSA | Acquired voxel size (mm) | Reconstruction matrix | Reconstructed voxel size (mm) | Scan duration (min) |
|---|---|---|---|---|---|---|---|---|---|
| $^{31}$P-Basal Ganglia | $^{31}$P 2D CSI ISIS localisation | On | 4000 | 0.22 | 10 | 40×40×20 | 12×12 | 17.5×17.5×20 | 12:48 |
| $^{31}$P-Substantia Nigra | $^{31}$P 2D CSI ISIS localisation | On | 4000 | 0.22 | 8 | 40×40×20 | 14×14 | 15×15×20 | 10:16 |

CSI, chemical shift imaging; ISIS, image-selected in vivo spectroscopy; NOE, nuclear overhauser efect; NSA, number of signal averages; TE, time to echo; TR, time to repetition.

subject's feet during walking, and various spatiotemporal gait parameters such as step time, stride length, step width and stance time are computed. The system has a spatial resolution of 1 cm and a temporal resolution of milliseconds. The data from the inertial sensors will be used to monitor truncal sway during walking and provide a set of additional digitally mobility outcomes associated to the quality of gait (eg, gait smoothness, variability, symmetry, etc).[38 39] The sensors will be positioned at both ankles, the lower back (L5), upper back (C7) and forehead. Each sensor contains an accelerometer, gyroscope and magnetometer and records synchronised data wirelessly. Data will be analysed with previously published, validated state of the art algorithms, implemented in Matlab.[38 40 41]

## STATISTICAL ANALYSIS

These analyses will include all randomised patients (an intention-to-treat analysis population). The primary outcome of safety and tolerability will be reported using descriptive statistics and comparisons between treatment groups. Demographic and clinical assessment data will be summarised.

For each of the secondary outcomes, the change from baseline will be summarised within treatment groups using standard summary statistics (number of participants, mean, SD, median, minimum and maximum) with appropriate statistical tests assessing for significant differences depending on the distribution of the data and any relevant covariates.

## DATA MANAGEMENT

Data will be kept in accordance with Good Clinical Practice, the Data Protection Act 2018 and General Data Protection Regulations. Data management will be provided by the University of Sheffield CTRU. All data will be entered remotely on to a centralised database held within the CTRU (Prospect) by a research study member at the study site. Access to Prospect is controlled by usernames and encrypted passwords.

All participants will be assigned a unique participant ID number at screening that will link all of the clinical information held about them on the study database. It will also be used in all correspondence between CTRU and participating centres.

## ETHICS AND DISSEMINATION

This trial has been approved by the East of England – Cambridgeshire and Hertford Shire Research Ethics committee (Protocol ID: 18/EE/0280) in November 2018. The study will be conducted in accordance with the local R&D approval and the Declaration of Helsinki. All participants provide written informed consent prior to any study procedures commencing. The results will be published in a peer-reviewed journal and presented at regional, national and international scientific meetings as appropriate. A plain English summary of the study results will be sent to the study participants once data analysis has been completed. Results of the study may also be presented at meetings of PD support groups or to other relevant lay audiences.

## DISCUSSION

We propose a novel study design for early, proof of concept PD neuroprotection trials, combining assessment for safety and tolerability with $^{31}$P-MRS-based assessment of target engagement of bioenergetics pathways and motion sensor-based objective quantification of disease progression. Our study protocol will be particularly powerful for any compound aiming to directly improve mitochondrial function in PD. Additionally, our approach of using $^{31}$P-MRS also holds promise to determine biologically relevant target engagement for compounds aiming at genetically defined upstream targets such as antisense oligonucleotides for *LRRK2* or antibody therapy for alpha-synuclein. Mitochondrial dysfunction is a well-recognised aspect of both LRRK2- and alpha-synuclein-associated PD.[42 43]

A recent open-label study of UDCA over 6 weeks with an escalating dose up to 50 mg/kg in five patients with mild to moderate PD found reasonable tolerability and also used $^{31}$P-MRS to assess target engagement.[44] However, their $^{31}$P-MRS imaging data were obtained in only three participants and their methodology differed in that a surface coil was used and to acquire occipital lobe spectra only.

In-depth sensor-based gait analysis has the potential to overcome the current limitations of the MDS-UPDRS-based clinical assessment.[18] Gait analysis provides a method of quantifying gait disability and postural instability and therefore has potential as an objective motor endpoint for future studies. There is clear evidence that greater axial involvement predicts a poorer outcome in PD with regard to both cognitive decline and postural instability.[23] It is, therefore, likely that the greatest value in sensor-based analysis is in assessing a combination of spatiotemporal and upper body gait characteristics both in the formal clinical setting but also in exploring real-life mobility through at-home monitoring.[38 45 46]

UDCA has previously been trialled in another neurodegenerative disorder, MND at doses of 15, 30 and 50 mg/kg in a total of 18 patients. Patients were treated for 4 weeks. The main AEs were minor gastrointestinal side effects, graded as mild to moderate. Side effect profiles and frequency were broadly similar between groups without a clear dose correlation.[13] This represents grounds to hypothesise that the primary outcome of safety and tolerability of UDCA at 30 mg/kg in PD will be achievable. We expect completion of the study analysis by July 2021.

**Author affiliations**
[1]Sheffield Institute for Translational Neuroscience, The University of Sheffield, Sheffield, UK
[2]NIHR Sheffield Biomedical Research Centre, Royal Hallamshire Hospital, Sheffield, UK

[3]Institute for In Silico Medicine, The University of Sheffield, Sheffield, UK
[4]Academic Unit of Radiology, The University of Sheffield, Sheffield, UK
[5]Department of Clinical and Movement Neurosciences, University College London Institute of Neurology, London, UK
[6]Statistical Services Unit, The University of Sheffield, Sheffield, UK

**Contributors** OB is responsible for the overall trial design with contributions from TF. SM led the overall administration and preparation of the trial. TF, SM and MA deliver the trial at the UCLH site. TP, MS, AA, NH, IDW and TJ are responsible for the implementation and analysis of the 31P-MRS. EB, AM and CM are responsible for the implementation and analysis of the sensor-based movement analysis tools. RT is responsible for statistical support of the trial and the power calculations provided. TP and EB are responsible for preparing the manuscript under the supervision of OB. All authors have reviewed and commented on this paper.

**Funding** This research was cofunded by the NIHR Sheffield BRC and the JP Moulton Charitable Foundation and was carried out at/supported by the NIHR Sheffield Clinical Research Facility. TP is funded by the NIHR Sheffield BRC and The Cure Parkinson's Trust. AA is funded by the Sheffield NIHR Sheffield BRC.

**Competing interests** None declared.

**Patient and public involvement** Patients and/or the public were involved in the design, or conduct, or reporting, or dissemination plans of this research. Refer to the Methods section for further details.

**Patient consent for publication** Not required.

**Provenance and peer review** Not commissioned; externally peer reviewed.

**ORCID iDs**
Thomas Payne http://orcid.org/0000-0001-6753-7847
Ellen Buckley http://orcid.org/0000-0002-0968-6286
O Bandmann http://orcid.org/0000-0003-3149-0252

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
