## [Reviewer comments · BMJ Open]

ARTICLE DETAILS

TITLE (PROVISIONAL)	The UP Study – Ursodeoxycholic acid as a novel disease-modifying treatment for Parkinson’s disease: Protocol for a two-centre, randomized, double-blind, placebo-controlled trial
AUTHORS	Payne, Thomas; Sassani, Matilde; Buckley, Ellen; Moll, Sarah; Anton, Adriana; Appleby, Matthew; Maru, Seema; Taylor, Rosie; McNeill, Alisdair; Hoggard, N; Mazza, Claudia; Wilkinson, Iain; Jenkins, Thomas; Foltynie, Thomas; Bandmann, O

VERSION 1 – REVIEW

REVIEWER	Dr Paul Chazot FBPhS Durham University UK
REVIEW RETURNED	23-Apr-2020

GENERAL COMMENTS	Within the scope of my expertise, this appears to be a clear, detailed and appropriate description, by an experienced group, of a proposed clinical trial protocol for an interesting novel drug treatment for Parkinson's disease, including a novel set of methods to evaluate primary and secondary outcomes. Perhaps some additional detail related to the mitochondria mechanism(s) for the drug for the non-specialist reader. There are one or two minor grammatical errors.
---

REVIEWER	Seyed-Mohammad Fereshtehnejad Division of Neurology, Department of Medicine, University of Ottawa, Ottawa, ON, Canada
REVIEW RETURNED	03-May-2020

GENERAL COMMENTS	[ ] INTRODUCTION 1. Please provide further explanation on how UDCA might have mitochondrial rescue actions. What is the pharmacologic role of UDCA in mitochondria? This needs further elaboration in the 'Introduction' section. [ ] METHODS 2. While it is reasonable to have a rather small number of patients (n=30) in the intervention arm of a phase II trial, I wonder why the placebo group is even smaller (n=10). In order to increase statistical power, I suggest increasing the number of patients in the placebo group to at least 30 (1:1 ratio). 3. Please explain your strategy to deal with lost-to-follow up? Is there any increase in sample size of each study arm to compensate for possible lost to follow-up cases? 4. The common dose of UDCA is 10-15 mg/kg/day; why authors decided to design this trial giving a higher dose of 30 mg/kg?
--

	Given that this is a phase II safety trial, why a regular dose of 15 mg/kg was not chosen? 5. Parkinson's disease is now known to have a long period of prodromal stage, as long as 20 years, prior to the time of diagnosis. Like other neurodegenerative diseases, clinically apparent stage might be quite late for neuroprotective / disease-modifying agents to show tangible effects on the residual neuronal reserves. I wonder why such a trial has not been designed for individuals with prodromal stage of Parkinson's disease such as REM sleep behavior disorders (RBD). Why do authors decide to recruit early PD and not people with prodromal parkinsonism? 6. Besides the combination of a large list of non-motor (e.g. RBD, dysautonomia, constipation, hyposmia, mood disorders, mild cognitive impairment) features, the motor phenotype and rate of progression are also quite variable from person to person in Parkinson's disease. Thus, patients with Parkinson's disease are now categorized into various subtypes (e.g. tremor-dominant vs. PIGD-dominant or purely motor predominant vs. diffuse malignant subtype with clearly different rate of progression. This should be considered in subjects' selections particularly in small sample size pilot trials. I highly suggest authors to include PD subtypes as an important variable, preferably try to balance distribution of subtypes between the two arms of the trial. 7. Please provide information whether the sensor-based devices / methods for quantification of the motor features of PD have been validated previously? 8. Biliary obstruction is listed as a contraindication for ursodiol. I wonder if this can be added to the list of exclusion criteria in Table 1. 9. Further information is needed on the details of statistical plans and procedures in the 'Statistical Analysis' section. For instance, please explain what exact statistical tests will be used to compare longitudinal numeric outcomes between the two groups. 10. How do authors deal with the routine treatment plans in these Parkinson's disease cohorts (i.e. levodopa, dopamine agonist, etc.)? Is there any strategy to harmonize dopaminergic treatment plans between the two study arms? As we know, there is considerable discrepancy between neurologist on how to start and continue PD treatment. Please elaborate on this very important issue as these medications can definitely affect motor and non-motor manifestations during the follow-up period. 11. In the section 'Ethics and Dissemination', please clarify whether participants will provide a written informed consent.
--	--

VERSION 1 – AUTHOR RESPONSE

Please find below our responses to the reviewers comments made regarding manuscript titled “**The UP Study – Ursodeoxycholic acid as a novel disease modifying treatment for Parkinson's disease: Protocol for a two-centre, randomized, double-blind, placebo-controlled trial**” with the manuscript ID of bmjopen-2020-038911.

We hope that these responses satisfy the comments made upon the work.

INTRODUCTION

1. Please provide further explanation on how UDCA might have mitochondrial rescue actions. What is the pharmacologic role of UDCA in mitochondria? This needs further elaboration in the 'Introduction' section.

We have added the below text to the manuscript to expand further on this.

The mode of action of UDCA remains to be fully elucidated. Current literature would suggest that it appears to be Akt mediated. Both Ursocholic acid and TUDCA have been demonstrated to induce Akt phosphorylation^{1 2} Akt activation requires phosphorylation at two sites and promotes cell survival through several mechanisms, failure of activation is a common finding underlying neurodegeneration². Reduced Akt signalling has been found in in-vitro models of PD and in sporadic PD brains post-mortem in the substantia nigra^{3 4}.

1. Castro-Caldas M, Carvalho AN, Rodrigues E, et al. Tauroursodeoxycholic acid prevents MPTP-induced dopaminergic cell death in a mouse model of Parkinson's disease. *Mol Neurobiol* 2012;46(2):475-86. doi: 10.1007/s12035-012-8295-4 [published Online First: 2012/07/10]
2. Mortiboys H, Aasly J, Bandmann O. Ursocholic acid rescues mitochondrial function in common forms of familial Parkinson's disease. *Brain* 2013;136(Pt 10):3038-50. doi: 10.1093/brain/awt224 [published Online First: 2013/09/04]
3. Timmons S, Coakley MF, Moloney AM, et al. Akt signal transduction dysfunction in Parkinson's disease. *Neurosci Lett* 2009;467(1):30-5. doi: 10.1016/j.neulet.2009.09.055 [published Online First: 2009/10/06]
4. Malagelada C, Jin ZH, Greene LA. RTP801 is induced in Parkinson's disease and mediates neuron death by inhibiting Akt phosphorylation/activation. *J Neurosci* 2008;28(53):14363-71. doi: 10.1523/JNEUROSCI.3928-08.2008 [published Online First: 2009/01/02]

METHODS

2. While it is reasonable to have a rather small number of patients (n=30) in the intervention arm of a phase II trial, I wonder why the placebo group is even smaller (n=10). In order to increase statistical power, I suggest increasing the number of patients in the placebo group to at least 30 (1:1 ratio).

Whilst we acknowledge that for any study assessing the disease modifying effect of a neuroprotective agent sample size and power is of critical importance we would like to highlight and clarify a few aspects of the study protocol presented.

- Our protocol presents two study arms, drug (n=20) and placebo (n=10), not of an n=30 in the intervention arm as clearly stated in the main body of the manuscript.
- The primary outcome of this study is safety and tolerability, the sample size and power has been calculated to investigate this outcome first and foremost. The calculations were performed using data from a similar repurposing trial investigating Exenatide in PD, which also had a primary outcome of safety and tolerability. See the section below extracted from the main text:
 - *“Should this study result in no SAEs then it would be of interest to determine how likely it is that a larger study would find an intolerable rate of SAEs. For this purpose, we will consider the rate of SAEs reported in the Exenatide PD trial to be tolerable and acceptable (i.e. 20%). In this study, should no SAEs be found in the group receiving UDCA (n=20) then the likelihood that the true SAE rate is less than 20% is 0.990778.”*
- The 2:1 ratio of drug to placebo ensures that the study can satisfy assessment for the primary outcome while also allowing the collection of exploratory data regarding the secondary outcomes which are assessing for the disease modifying effect of UDCA. We would like to emphasise that the data collected on secondary outcomes is to allow an estimate of the effect size and variance of UDCA on motor severity using standard clinical assessment tools, ³¹P-MRS measures of mitochondrial function and the objective quantification of motor impairment using a sensor-based approach for future trial design. We have added the following to the main text to clarify this:
 - *The data collected for the secondary and exploratory outcomes will allow the estimation of the effect size and variance in each outcome to facilitate formal power calculations for future Phase III studies. Of note, there is currently no data using either 31P-MRS or our sensor based approached quantification of motor impairment.*

The collection of such data is critical to allow high quality future trial design using these novel outcome measures.

- The trial has already commenced and therefore revision of sample sizes will not be possible.

3. Please explain your strategy to deal with lost-to-follow up? Is there any increase in sample size of each study arm to compensate for possible lost to follow-up cases?

We thank the reviewer for this comment, we have added the following text to the Sample size section to explain our approach:

The sample size has not been prospectively adjusted to account for any loss to follow-up. Instead, as the trial is of a relatively short duration we have instead allowed for any participants withdrawing from the study or lost to follow-up before the completion of 12 weeks of treatment to be replaced with a new participant.

4. The common dose of UDCA is 10-15 mg/kg/day; why authors decided to design this trial giving a higher dose of 30 mg/kg? Given that this is a phase II safety trial, why a regular dose of 15 mg/kg was not chosen?

Previous work done in Motor Neuron Disease has led our decision to trial 30mg/kg as stated in the main text. Our choice of dose was based upon preliminary experiments assessing the EC90 of UDCA using in vitro neuronal models, we found the EC90 of UDCA was 100 nM in most of our experiments, this is currently unpublished data. However, we have added the below text to the Design section to state the following:

The choice of 30mg/kg day has been informed by previous pharmacokinetic studies in Motor Neuron Disease, this dose allows effective penetrance of the CNS but also balances the exposure to a potentially higher risk of side effects with increasing doses and possible issues with compliance due to the then very large number of additional tablets the patients would need to take.

5. Parkinson's disease is now known to have a long period of prodromal stage, as long as 20 years, prior to the time of diagnosis. Like other neurodegenerative diseases, clinically apparent stage might be quite late for neuroprotective / disease-modifying agents to show tangible effects on the residual neuronal reserves. I wonder why such a trial has not been designed for individuals with prodromal stage of Parkinson's disease such as REM sleep behavior disorders (RBD). Why do authors decide to recruit early PD and not people with prodromal parkinsonism?

REM sleep behaviour disorder (RBD) is well established to be a potentially prodromal phase to the development of PD, however the time from diagnosis of RBD to conversion to PD is highly variable and potentially long. A large recent multicentre study determining predictors of conversion of RBD to both dementia and parkinsonism would suggest that the rate of conversion is around 6.3% per year in RBD. Sample size calculations provided by the same study would suggest a sample size of 366 participants per intervention arm (a total of 732) for a 2 year neuroprotective trial assessing all non-stratified RBD. Even stratifying and including only those RBD cases with strong predictors of conversion to parkinsonism and dementia reduces the required sample size to around 200 participants per intervention arm (around 400 total).

For reference: Postuma RB, Iranzo A, Hu M, et al. Risk and predictors of dementia and parkinsonism in idiopathic REM sleep behaviour disorder: a multicentre study. *Brain*. 2019;142(3):744-759. doi:10.1093/brain/awz030

6. Besides the combination of a large list of non-motor (e.g. RBD, dysautonomia, constipation, hyposmia, mood disorders, mild cognitive impairment) features, the motor phenotype and rate of progression are also quite variable from person to person in Parkinson's disease. Thus, patients with Parkinson's disease are now categorized into various subtypes (e.g. tremordominant vs. PIGD-dominant or purely motor predominant vs. diffuse malignant subtype with clearly different rate of progression. This should be considered in subjects' selections particularly in small sample size pilot trials. I highly suggest authors to include PD subtypes as an important variable, preferably try to balance distribution of subtypes between the two arms of the trial.

We agree with the reviewer that motor heterogeneity is a particular challenge in clinical trials in PD and that tools to assess for risk of progression in particular are of key importance. Although not formally mentioned in the manuscript we will be calculating a validated risk of progression score validated across two large population cohorts as an exploratory variable. We have included the following text to clarify this:

As an additional variable to be used in exploratory analysis a validated prognostic model calculating the risk of progression to an unfavourable outcome (either postural instability or dementia at 5 years) will be applied to each participant. We hope that this variable will account for some of the inherent heterogeneity among participants for their speed of clinical progression.

References

Velseboer, D. C., R. M. de Bie, L. Wieske, J. R. Evans, S. L. Mason, T. Foltynie, B. Schmand, R. J. de Haan, B. Post, R. A. Barker and C. H. Williams-Gray (2016). "Development and external validation of a prognostic model in newly diagnosed Parkinson disease." Neurology **86**(11): 986-993.

We feel that formally subtyping each participant according to tremor dominant and PIGD-dominant PD will add additional confounders to data analysis that the trial has not been designed to account for as this would require larger numbers to appropriately balance and compare between subgroups. Additionally, there is a great deal of variance between rates of progression based on these subtypings and there is also significant conversion between clinical subtypes. There remains a paucity of data on the true differences in progression between subtypes however more recent data suggest that although PIGD-PD have more severe clinical manifestations early in the disease, the main differences in progression are in neuropsychiatric features, rather than in motor progression as modelled from the PPMI dataset.

Reference:

Simuni, T., C. Caspell-Garcia, C. Coffey, S. Lasch, C. Tanner, K. Marek and P. Investigators (2016). "How stable are Parkinson's disease subtypes in de novo patients: Analysis of the PPMI cohort?" Parkinsonism Relat Disord **28**: 62-67.

Aleksovski, D., D. Miljkovic, D. Bravi and A. Antonini (2018). "Disease progression in Parkinson subtypes: the PPMI dataset." Neurol Sci **39**(11): 1971-1976.

7. Please provide information whether the sensor-based devices / methods for quantification of the motor features of PD have been validated previously?

The previous literature published validating these techniques and algorithms has all been referenced in the main text. Although we acknowledge that providing further information in the body of the main text may be desirable, the technical detail required to truly relay these techniques is out of the scope of a protocol paper, we have therefore left the text as it currently is but amended the final sentence of the paragraph:

Data will be analysed with previously published, validated state of the art algorithms, implemented in Matlab.

8. Biliary obstruction is listed as a contraindication for ursodiol. I wonder if this can be added to the list of exclusion criteria in Table 1.

We have now added this to Table 1.

9. Further information is needed on the details of statistical plans and procedures in the 'Statistical Analysis' section. For instance, please explain what exact statistical tests will be used to compare longitudinal numeric outcomes between the two groups.

As many of the secondary outcomes are examining outcome measures not previously utilised in clinical trials (other than the MDS-UPDRS), we are unable to be more specific with the exact statistical test as the distribution and nature of the data is not currently known and therefore an appropriate statistical test cannot be chosen at this time.

10. How do authors deal with the routine treatment plans in these Parkinson's disease cohorts (i.e. levodopa, dopamine agonist, etc.)? Is there any strategy to harmonize dopaminergic treatment plans between the two study arms? As we know, there is considerable discrepancy between neurologist on how to start and continue PD treatment. Please elaborate on this very important issue as these medications can definitely affect motor and non-motor manifestations during the follow-up period.

As the primary outcome is of safety and tolerability and not for the assessment of disease modification we felt that mandating specific treatment plans was not ethically justified. Given the motor heterogeneity of PD, many PD patients require varying treatment plans to adequately control their motor symptoms and we did not feel that in a Phase II study with this primary outcome in mind that potentially compromising their current treatment was justified. All participants are managed by Movement Disorders Specialists and therefore receive high quality routine clinical care. We are attempting to account for this by calculating the total levodopa equivalent daily dosage for each participant to be used as a covariate in relevant statistical analyses, this outcome is listed in the exploratory outcomes section.

11. In the section 'Ethics and Dissemination', please clarify whether participants will provide a written informed consent.

We have amended this accordingly.

VERSION 2 – REVIEW

REVIEWER	Seyed-Mohammad Fereshtehnejad University of Ottawa, Ottawa, ON, Canada
REVIEW RETURNED	18-Jun-2020
GENERAL COMMENTS	All my comments and questions have been adequately addressed or answered.